# Overexpression of *MEKK18* from *Arabidopsis pumila* in rice significantly enhances stress resistance at the early stage

Junrong Li[1☯], Yilin Yang[1,2☯], Zhanglun Sun[1,2☯], Edwin Afriyie Owusu[1], Leiming Liu[1,2], Tianrun Mei[1], Qingli Zhang[1], Xianzhong Huang [1*]

1 Key Laboratory of Crop Germplasm Innovation and Green Production, College of Agriculture, Anhui Science and Technology University, Chuzhou Anhui, China, 2 College of Life Sciences, Shihezi University, Shihezi, Xinjiang, China

☯ These authors contributed equally to this work.
* huangxz@ahstu.edu.cn

## Abstract

The mitogen-activated protein kinase (MAPK) signaling cascade is a crucial signaling component of eukaryotic cells. It senses endogenous or exogenous stimuli and exerts a critical regulatory influence on stress responses. The *MAP kinase kinase kinase* (*MEKK*) gene is located upstream in the MAPK cascade. It is extensively engaged in plant growth and development, playing a pivotal role in tolerance to abiotic stresses such as salt, drought, and extreme temperatures. *Arabidopsis pumila*, a short-lived cruciferous plant native to the southern desert area of the Gurbantünggüt Desert in Xinjiang exhibits excellent stress adaptation and contains a rich array of stress resistance genes. Previous research has demonstrated that the expression of *ApMEKK18* is continuously upregulated under saline stress, although its function in the response to this type of abiotic stress is unclear. The findings of this study reveal that the ApMEKK18 protein is localized in the cell nucleus, and the *ApMEKK18* gene is upregulated in response to abscisic acid (ABA), NaCl, PEG6000, and mannitol, exhibiting varying expression patterns under different conditions. Using *Agrobacterium*-mediated transformation, we overexpressed *ApMEKK18* in rice. Compared to the control variety Nipponbare (NP), no substantial differences were observed in plant height, number of primary and secondary branches, grain width, and thickness in the *ApMEKK18*-overexpressing transgenic rice. However, the number of tillers, panicle length, grain length, 1,000-grain weight, and single-plant yield were significantly increased. Furthermore, *ApMEKK18* overexpression enhanced seed germination under high salt and ABA stress conditions, while reducing sensitivity to exogenous ABA and improving salt tolerance in seedlings. The results of this study provide a theoretical foundation for further research on the function of *ApMEKK18* and lay the groundwork for mining stress-resistance genes from *A. pumila*. Additionally, the findings offer insights into improving crop stress tolerance through genetic engineering.

**Data availability statement:** All relevant data are within the paper and its Supporting Information files.

**Funding:** This work was supported by the National Natural Science Foundation of China (32270385), the Excellent Scientific Research and Innovation team of the Education Department of Anhui Province (2022AH010087), the Science and technology innovation team of Anhui Sciences and Technology University (2023KJCXTD001), the Talent Introduction Start-up Fund Project of Anhui Science and Technology University (NXYJ202001), the Construction Funds for Crop Science of Anhui Science and Technology University (XK-XJGF001), and the National Innovation and Entrepreneurship Training Program for College Students (202310879009). The funders had no role in study design, data collection and analysis, decision to publish, or preparation of the manuscript.

**Competing interests:** The authors have declared that no competing interests exist.

## Introduction

Abiotic stresses such as salinity, drought, high temperatures, nutrient deficiency, and heavy metals in the soil can disrupt the stability of plant growth during daily and seasonal cycles, leading to changes in physiological status [1,2]. Extreme climate variations result in reduced crop yields and quality while also exacerbating the adverse impacts of abiotic stresses on agricultural production [3]. Currently, over one-third of irrigated land in China is threatened by salinization, which reduces water-use efficiency and leads to the excessive accumulation of $Na^+$ and $Cl^-$ in plants, severely affecting the sustainable development of modern agriculture [4,5].

Drought and saline stress cause physiological damage to plants, affecting their metabolism, inhibiting growth and development, and negatively impacting both yield and quality [6,7]. Under stress conditions, plants must readjust their physiological and biochemical responses and ion dynamics to mitigate secondary damage, enabling them to adapt to the adverse environment [8,9]. In response to drought and saline stress, plants can maintain cell volume and osmotic pressure through osmotic adjustment [6], an adaptive mechanism involving the aggregation of soluble substances, such as proline, soluble sugars, and betaine, which promotes water absorption. This biochemical response is strongly dependent on the osmotic level of water [10]. The synthesis of proline effectively prevents protein moisture loss and denaturation caused by osmotic stress and helps to eliminate excess reactive oxygen species (ROS) [11,12]. Betaine is a crucial osmotic regulator that critically contributes to the maintenance of osmotic pressure. Betaine application can alleviate damage to maize seedlings under drought [13]. Soluble sugars, including glucose, fructans, sucrose, and starch, accumulate under drought and saline stress and play a role in osmotic regulation, thereby enhancing tolerance to these abiotic stresses [14–16]. Although excess $Na^+$, $Cl^-$, and $K^+$ ions can be toxic to plants, they act as regulators in salt-tolerant plants, helping to maintain the osmotic balance [14,17]. To remove ROS generated under salt stress, plants have evolved an antioxidant system comprising both enzymatic and non-enzymatic components to protect cells from oxidative damage [18]. The enzymatic system includes Catalase (CAT), Ascorbate peroxidase (APX), Peroxidase (POD), Superoxide dismutase (SOD), Glutathione peroxidase (GR), Dehydroascorbate reductase (DHAR), and Monodehydroascorbate reductase (MDHAR) [19]. For example, overexpressing the *CBL-Interacting Protein Kinase 6a* (*GhCIPK6a*) gene in cotton (*Gossypium hirsutum*) enhances the activities of SOD and POD, reduces the accumulation of hydrogen peroxide ($H_2O_2$), and defends seedlings against membrane impairment under saline stress [20]. In tomatoes (*Solanum lycopersicum*), overexpression of the *Prunus persica* SNF-related Kinase 1 (*PpSnRK1*) gene induces the expression of relevant antioxidant genes, clearing ROS accumulation and improving salt tolerance [21].

The mitogen-activated protein kinase (MAPK) signaling cascade is an important signaling module in eukaryotic cells, whereby receptor-mediated signals are transduced into cellular responses, participating in various biological and abiotic stress responses, plant hormone signaling events, and processes such as cell division and development [22–26]. The MAPK cascade is mainly composed of three types

of enzymes: MAP kinase kinase kinase (MAPKKK or MEKK), MAP kinase kinase (MAPKK or MEK), and MAPK, which sequentially activate each other through phosphorylation [22]. MEKK is a serine/threonine kinase that can phosphorylate the S/T-X3–5-S/T motif in the activation loop of MEK. MEK activates the dual-specificity kinase MAPK by phosphorylating of the T-X-Y motif in its activation loop. Notably, MEKK, at the core of the MAPK cascade, can receive upstream signals [27]. The first MAPK signaling pathway (MEKK1-MKK4/5-MPK3/6 cascade) involved in plant innate immune defense was discovered in *Arabidopsis thaliana* [28,29]. Studies have shown that the MAPK cascade constitutes a highly pre-served signaling mechanism in higher plants [22]. Overexpression of the *AtMEKK16* gene in *Arabidopsis* seeds increases sensitivity to abscisic acid (ABA) during germination and relieves the inhibitory effects of ABA on root development. It has been found that the MEKK16 kinase acts as a negative regulator, critically influencing seed germination, root growth, and cotyledon chlorophyll synthesis [30].

The *Arabidopsis* genome contains at least 80 *MEKK* genes, 10 *MEK* genes (*MAPKK1-MAPKK10*), and 20 *MAPK* genes (*MAPK1-MAPK20*) [27]. However, only a few MEKKs regulate ABA or other related signaling pathways to facilitate the plant's prompt reaction to stressful environments. Evidence has revealed that MEKK18 plays a key regulatory role in ABA-mediated responses to abiotic stress, interacting with downstream protein kinases in the MAPK cascade, such as MEK3, MAPK1/2/7/14, and MEKK18, which are involved in ABA signaling, leaf senescence, and drought resistance [25,31]. *Arabidopsis* ABA-responsive binding factors (ABF2, ABF3, and ABF4) can bind to the ABA-responsive *cis*-acting elements in the promoter of the *MEKK18* gene, transactivating its expression and exerting a critical regulatory influence on the initiation of leaf senescence [26]. *MEKK18* gene expression is significantly induced under drought stress and ABA treatment [32]. The MEK1-MPK6 cascade regulates the expression of the *Catalase 1* (*CAT1*) gene in the ABA signaling pathway, leading to the production of $H_2O_2$ [33]; MEK3-MPK1/2 mediates ABA signaling and participates in salt stress tolerance; and MPK9/12, through ROS-mediated ABA signaling, can positively regulate stomatal closure [34]. These func-tions demonstrate that the MAPK cascade plays a crucial role in plant growth, development, and stress responses, with many stress signals transduced into cellular responses at the molecular and physiological levels [35].

However, research on the MAPK cascade pathway in response to abiotic stress has mainly focused on *MEK* and *MAPK* genes, while studies on the upstream *MEKK* gene family and its functions in the MAPK cascade are relatively limited. As a result, the discovery of stress resistance-related genes in the MAPK cascade pathway has been limited [35]. Therefore, the regulatory network involving *MEKK* gene family transcription in response to diverse stresses requires further investigation.

*Arabidopsis pumila*, which grows primarily in the southern desert areas of the Gurbantünggüt Desert in Xinjiang, is a cruciferous early-spring ephemeral plant with characteristics such as a short growth cycle, high seed yield, high repro-ductive rate, and strong stress resistance [8,36–38]. Mining stress resistance-related genes from the genome of *A. pumila* and exploring the mechanisms by which they respond to desert environments can help to elucidate the molecular mechanisms behind the plant's rapid flowering and maturation rates under desert conditions. In a study by Yang et al. [8], RNA-sequencing analysis of *A. pumila* revealed that a *MEKK18* homolog (S1 Fig.) was consistently upregulated under continuous salt stress, suggesting that the gene plays a key regulatory role under saline conditions. In this study, we examined the expression of *ApMEKK18* in response to various abiotic stresses, such as ABA, salt, and drought, and over-expressed the gene in rice. The results showed that the cross-species overexpression of *ApMEKK18* improves tolerance to drought, salt, and ABA stress in rice. This study provides a theoretical foundation for further investigating *ApMEKK18* function. It lays the groundwork for identifying stress-resistance genes in *A. pumila*, offering insights into genetic engineer-ing strategies for enhancing stress resistance in crops.

## Materials and methods

### Plant materials and growth conditions

*Arabidopsis pumila* seeds from Xinjiang were stored in our laboratory [8]. The seeds were surface-sterilized and sown on 1/2 MS medium.

The seeds were then cultivated in an incubator (16 hours light/8 hours dark, 22°C, 200 μmol m$^{-2}$ s$^{-1}$). Samples of different tissues were collected at various stages: cotyledons (8 days), hypocotyl (8 days), young roots (8 days), petioles (30 days), adventitious roots (30 days), and rosette leaves (30 days). Seven-day-old seedlings exhibiting normal growth were planted in soil. Stems, leaves, and flower buds were collected after 60 days of growth, and flowers were harvested at 65 days. At maturity (70 days), fruit stems, pods, and seeds were collected. All samples were frozen rapidly in liquid nitrogen and preserved at –80°C.

To observe the effects of abiotic stress on the growth and development of *A. pumila*, surface-sterilized seeds were sown on 1/2 MS medium. After a 2-day cold treatment (vernalization) at 4°C, the seedlings were transferred to a long-day light-controlled incubator (16 hours light/8 hours dark, 22°C, 200 μmol m$^{-2}$ s$^{-1}$). Seven-day-old seedlings were transferred to 1/2 MS medium containing 1 μM ABA, and tissue samples were taken at 0, 1, 6, and 12 h. Additionally, 14-day-old seedlings were transferred to 1/2 MS medium containing 250 mM NaCl, 20% PEG6000, or 300 mM mannitol. Samples were taken at 0, 1, 6, and 12 h for gene expression analysis. The experiment was conducted independently three times, and the samples were stored at –80°C.

### Subcellular localization in *Arabidopsis* protoplasts

To amplify the target fragment, primers lacking a stop codon were designed based on the *ApMEKK18* [8] coding sequence (CDS) (S1 Table). The *ApMEKK18* CDS, which lacks a stop codon, was cloned into the *pEASY-Blunt-Zero* (TRAN, Beijing) intermediate vector. The construct was transformed into *Escherichia coli*, and the correctly sequenced gene fragment was ligated with the *pCAMBIA2300-35S:GFP* vector [39] using T$_4$ ligase and transformed into *E. coli* for plasmid extraction. The plasmid was verified by restriction enzyme digestion, and the product was introduced into *Agrobacterium tumefaciens* GV3101. The subcellular localization of the ApMEKK18-GFP fusion protein in *Arabidopsis* protoplasts was observed using a laser confocal microscope [40].

### Quantitative real-time PCR

Total RNA was extracted from samples using the FastPure Plant Total RNA Isolation Kit (Vazyme, Nanjing). The RNA was then reverse transcribed into cDNA using the HiScript III 1st Strand cDNA Synthesis Kit (+gDNA wiper) (Vazyme, Nanjing) according to the manufacturer's instructions. The cDNA products were used as templates for quantitative real-time PCR (qRT-PCR). Specific primers for amplifying the *ApMEKK18* gene were designed using Primer 5.0 (S1 Table), and *ApGAPDH* was used as the reference gene [38]. The qPCR amplification system was prepared according to the instructions for the MonScript™ ChemoHS Specificity Plus qPCR Mix (Low ROX) (Suzhou Mona Biotechnology). Each 10 μL of reaction mixture consisted of: 0.1 μL Low ROX Dye (100×), 5 μL MonAmp™ ChemoHS qPCR Mix, 1 μL (20 ng/μL) cDNA template, 0.2 μL of each upstream and downstream primer (10 μmol/L), and nuclease-free Water to the final volume. The thermal cycling program included an initial denaturation at 95°C for 30 seconds, followed by 40 cycles of 95°C for 10 seconds and 60°C for 30 seconds, with a melt curve step at 95°C for 15 seconds, 60°C for 60 seconds, and 95°C for 15 seconds. Each PCR reaction was performed in triplicate. PCR was conducted using the ABI ViiA7 Real-Time PCR System (Life Technologies, USA). The relative expression levels of genes in different tissues and under different salt stress conditions were evaluated using the $2^{-\Delta Ct}$ method [41].

### Construction of the *ApMEKK18* gene overexpression vector and genetic transformation of rice

*ApMEKK18* was previously identified as *MEKK18* homologous gene through RNA-seq analysis [8]. Phylogenetic analysis showed that *ApMEKK18* was most closely related to *AtMEKK18* (S1 Fig). Based on the *MEKK18* gene sequence reported by Yang et al. specific primers for amplifying the CDS were designed (S1 Table) [8]. The *ApMEKK18* gene was amplified using PCR with a reaction mixture containing 10 μL of 2×Phanta Max Master Mix, 0.5 μL of both upstream and downstream primers, 2 μL of cDNA from flower tissue, and ddH$_2$O to a final volume of

                                                                                    

20 µL. The PCR program was as follows: 95°C for 3 minutes; then 95°C for 15 seconds, 56°C for 15 seconds, and 72°C for 78 seconds for 35 cycles, followed by a final extension at 72°C for 10 minutes. The amplified products were analyzed by 1.5% agarose gel electrophoresis. The target fragment was recovered, ligated into the *pEASY* vector, and transformed into *E. coli DH5α*, which was cultured overnight at 37°C. Positive clones were selected, and plasmids were extracted and identified by restriction enzyme digestion. Sequencing was performed by Nanjing General Bioengineering Company (Chuzhou branch). After confirming the correct sequence, the plasmid was transferred into the *pCAMBIA2300-35S-OCS* vector containing the cauliflower mosaic virus-derived 35S promoter [39], resulting in the construction of the binary expression vector *35S:ApMEKK18*. The plasmid containing *35S:ApMEKK18* was then transformed into the rice variety Nipponbare (NP) using *Agrobacterium* EAH105. In accordance with the method detailed by Molina-Risco et al. [42], nine $T_0$ transgenic rice lines were obtained after induction, subculture, co-cultivation, selection, differentiation, and rooting. *ApMEKK18* overexpression DNA was extracted from the leaves of these transgenic rice plants for molecular identification, and $T_3$ overexpression lines were selected. RNA was extracted from the young leaves of the identified positive plants, reverse transcribed into cDNA, and qRT-PCR was conducted to evaluate gene expression levels, with *OsActin* as the reference gene [43] (S1 Table). The method used for qRT-PCR was the same as described above.

## Investigation of rice plant architecture and grain traits

After soaking seeds from NP and *ApMEKK18* overexpression plants in water at 37°C for 2 days to induce germination, they were planted in the experimental site at Anhui University of Science and Technology (32°87' N, 117°56' E). Once the plants matured, agronomic traits such as plant height, tiller number, panicle length, the number of primary branches, and the number of secondary branches were recorded. The 1 000-grain weight and single-plant yield were determined using an electronic balance (Shanghai Yuping, YP3002).

## ABA stress experiment

Rice Seed Germination Experiment: Seeds from NP and rice *ApMEKK18* overexpression lines were surface-sterilized with hydrogen peroxide and sodium hypochlorite [44]. After sterilization, the seeds were sown evenly on 1/2 MS plates containing 0, 2, or 4 µM ABA and cultured in a growth chamber (14 hours light/10 hours dark, 25°C, 200 µmol m$^{-2}$ s$^{-1}$) for 7 days. The germination rate was recorded once a day, with three replicates per treatment concentration and 50 seeds per plate.

Rice Seedling Growth Experiment: Seeds from NP and *ApMEKK18* transgenic rice lines were surface-sterilized with hydrogen peroxide and sodium hypochlorite and soaked in water at 37°C for 2 days to promote germination. The seedlings were then transplanted into a Yoshida rice nutrient solution [45] containing 0, 2, or 4 µM ABA and cultured for 7 days or 14 days. Photos were taken, and plant height was measured at 7 and 14 days.

## Salt stress treatment experiment

Rice Seed Germination Experiment: Seeds from NP and *ApMEKK18* transgenic rice lines were surface-sterilized with hydrogen peroxide and sodium hypochlorite. After sterilization, the seeds were sown evenly on 1/2 MS plates containing 0, 50, 100, and 150 mM NaCl. The plates were sealed with sealing film and arranged in a growth chamber (14 hours light/10 hours dark, 25°C, 200 µmol m$^{-2}$ s$^{-1}$) for 7 days. The germination rate was recorded once a day, with three replicates per NaCl concentration and 50 seeds per plate.

Rice Seedling Growth Experiment: Seeds from NP and *ApMEKK18* transgenic rice lines were surface-sterilized with hydrogen peroxide and sodium hypochlorite, then soaked in water at 37°C for 2 days to promote germination. The seedlings were transplanted into Yoshida nutrient solution and cultured for 30 days. They were then transferred to Yoshida nutrient solution containing 150 mM NaCl for 14 days of stress treatment and then back to normal nutrient solution for 7 days of recovery.

## Results

### Expression profile of the *ApMEKK18* gene in different tissues and under abiotic stress

*ApMEKK18* gene expression levels in 13 tissue types from *A. pumila* were analyzed using qRT-PCR. Flowers exhibited the highest expression, followed by the pedicel, flower buds, and young roots (S2 Fig). Lower expression levels were seen in mature roots, hypocotyls, and petioles, while *ApMEKK18* expression was nearly undetectable in the remaining tissue types.

To explore the role of *ApMEKK18* in the abiotic stress response, expression in seedlings treated with 1 μM ABA, 250 mM NaCl, 20% PEG6000, or 300 mM mannitol was analyzed using qRT-PCR. The results suggested that *ApMEKK18* was significantly upregulated under all stress conditions, although the expression patterns varied (Fig 1). Under 1 μM ABA stress for 1 and 6 hours, *ApMEKK18* was consistently upregulated; expression decreased at 12 hours but remained above that of the control during the entire stress period (Fig 1A). Under 250 mM NaCl stress, *ApMEKK18* was rapidly upregulated and highly expressed throughout the treatment period (Fig 1B). Under 20% PEG6000 stress, *ApMEKK18* was rapidly upregulated for 1 hour, then gradually downregulated, but expression remained higher than in the control (Fig 1C). Similarly, under 300 mM mannitol stress, *ApMEKK18* was rapidly upregulated for 1 hour, then gradually downregulated, exhibiting similar expression at 6 and 12 h after the start of the treatment. Expression remained higher than in the control throughout the treatment period (Fig 1D). In summary, all four abiotic stress treatments induced *ApMEKK18* gene expression, suggesting that *ApMEKK18* plays an important role in the response to abiotic stress.

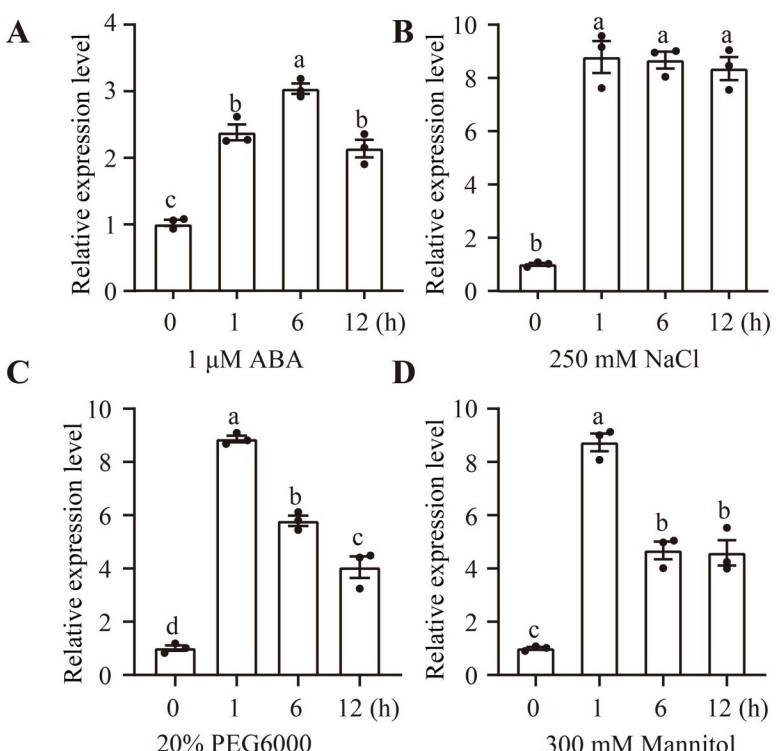

**Fig 1. Expression characteristics of *ApMEKK18* gene in response to four abiotic stresses.** Expression characteristics of the *ApMEKK18* gene in seedlings treated with 1 μM ABA **(A)**, 250 mM NaCl **(B)**, 20% PEG6000 **(C)**, and 300 mM Mannitol **(D)**, respectively. The Various lowercase letters represent statistically significant differences at $P < 0.05$. The statistical analysis was performed using one-way ANOVA and Duncan's multiple comparison test.

## Subcellular localization of ApMEKK18-GFP

To analyze the subcellular localization of the ApMEKK18 protein, a vector containing the *ApMEKK18* gene CDS lacking the stop codon (*35S:ApMEKK18-GFP*) or a control vector (*35S:GFP*) was introduced into *Arabidopsis thaliana* protoplasts. Laser scanning confocal microscopy revealed that in *Arabidopsis* protoplasts, the GFP signal from 35S:GFP was distributed throughout the entire cell (Fig 2), whereas the green fluorescence signal of the ApMEKK18-GFP fusion protein was mainly concentrated in the nucleus, indicating that ApMEKK18 is primarily a nucleus-localized protein.

## Generation of *ApMEKK18* overexpression transgenic rice

*Agrobacterium* strain EHA105 containing the *35S:ApMEKK18* vector was used to transform the rice variety NP via *Agrobacterium*-mediated transformation. After induction (S3A Fig), subculturing (S3B Fig), co-cultivation (S3C Fig), selection (S3D Fig), differentiation (S3E Fig), rooting (S3F Fig), and transplanting (S3G Fig), several $T_0$ transgenic rice lines were obtained. After multiple generations of breeding, nine homozygous transgenic lines were established. Genomic PCR analysis detected the target gene *ApMEKK18* in all transgenic plants (S4 Fig).

qRT-PCR was utilized to evaluate *ApMEKK18* expression in each of the transgenic rice lines. Compared to wild-type NP, the expression levels of *ApMEKK18* were similar in the *ApMEKK18*-OE1/OE2 lines, while the relative expression level in *ApMEKK18*-OE3 was upregulated ($P < 0.5$). Furthermore, expression was significantly upregulated in the other six transgenic lines ($P < 0.01$) (S5 Fig).

## Effect of *ApMEKK18* overexpression on rice plant architecture and grain morphology

There were no significant differences in plant height between the *ApMEKK18* overexpression lines and NP (Figs 3A and 3B). However, the number of tillers was significantly increased (Fig 3C). Panicle phenotype analysis (Fig 3D) and statistical results indicated that the average panicle length in *ApMEKK18*-OE7/OE8 was 2 cm longer than in NP (Fig 3E).

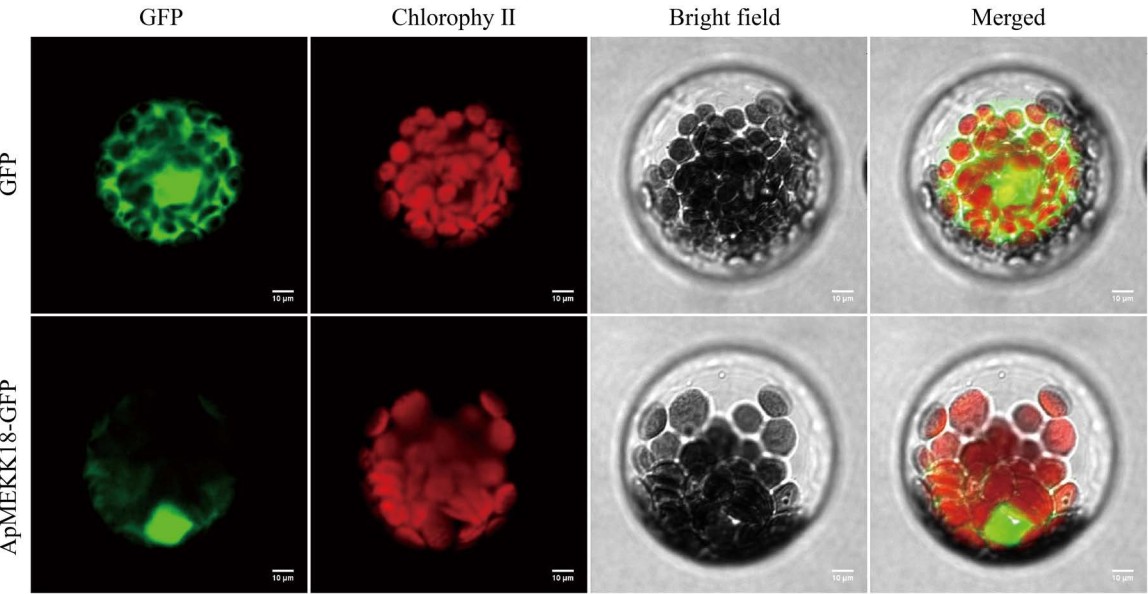

**Fig 2. Subcellular localization of ApMEKK18-GFP in *Arabidopsis* protoplasts following the introduction of 35S:GFP and 35S:ApMEKK18-GFP expression vectors.** Micrographs show cells expressing GFP and ApMEKK18-GFP fusion proteins. GFP signals were examined using fluorescence field illumination. Chlorophyll Red signals indicated chlorophyll autofluorescence. Scale bars = 10 μm.

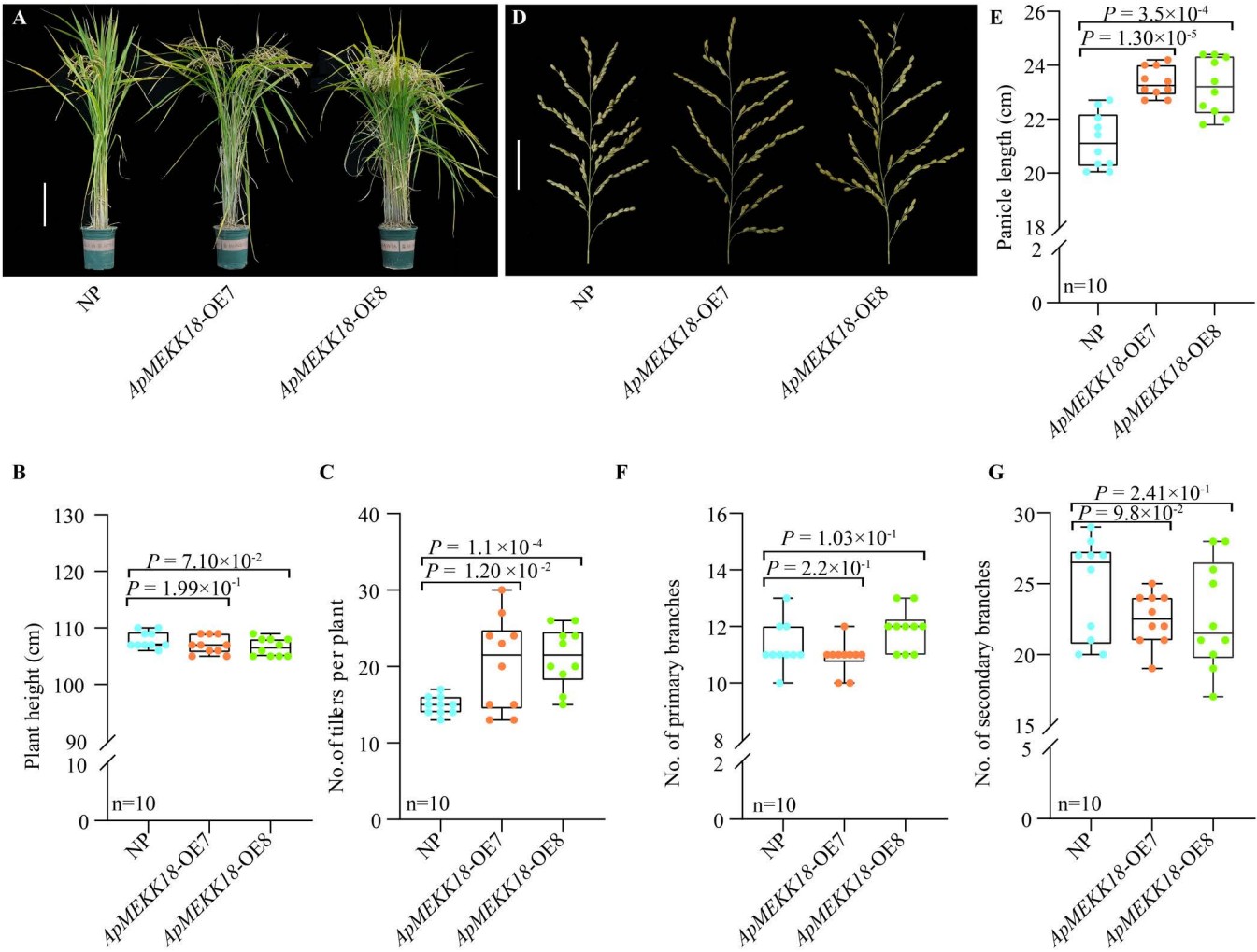

**Fig 3. Effects of *ApMEKK18* overexpression on plant growth and spike in rice. (A)** Plant growth under field growth conditions, scale bar: 20 cm. **(B)** Plant height. **(C)** Number of tillers. **(D)** Spike shape, scale bar: 5 cm. **(E)** Spike length. **(F)** Number of primary pedicels. **(G)** Number of secondary pedicels. NP: Nipponbare; OE-7 and OE-8 represent two independent *ApMEKK18* transgenic rice strains; one-way ANOVA (Student's *t*-test).

However, no significant variations were found in the number of primary branches (Fig 3F) or secondary branches (Fig 3G). Compared to NP, the grain length in *ApMEKK18*-OE7/OE8 was significantly increased (Figs 4A and 4B), while there were no meaningful differences in grain width (Figs 4C and 4D) and thickness (Figs 4E and 4F). Measurements of single-plant yield and 1,000-grain weight showed that overexpression of *ApMEKK18* significantly increased both single-plant yield (Figs 4G and 4H) and 1,000-grain weight (Fig 4I).

## Overexpression of *ApMEKK18* in rice increases the seed germination rate and reduces seedling sensitivity to exogenous ABA

To observe the effects of *ApMEKK18* overexpression on seed germination and seedling growth, NP and *ApMEKK18*-OE7/ OE8 seeds were treated with 0, 2, or 4 μM ABA, and germination rates were recorded over the next 7 days. No marked variations in germination rates between *ApMEKK18*-OE7/OE8 and NP seeds were seen without treatment (Fig 5A). Under

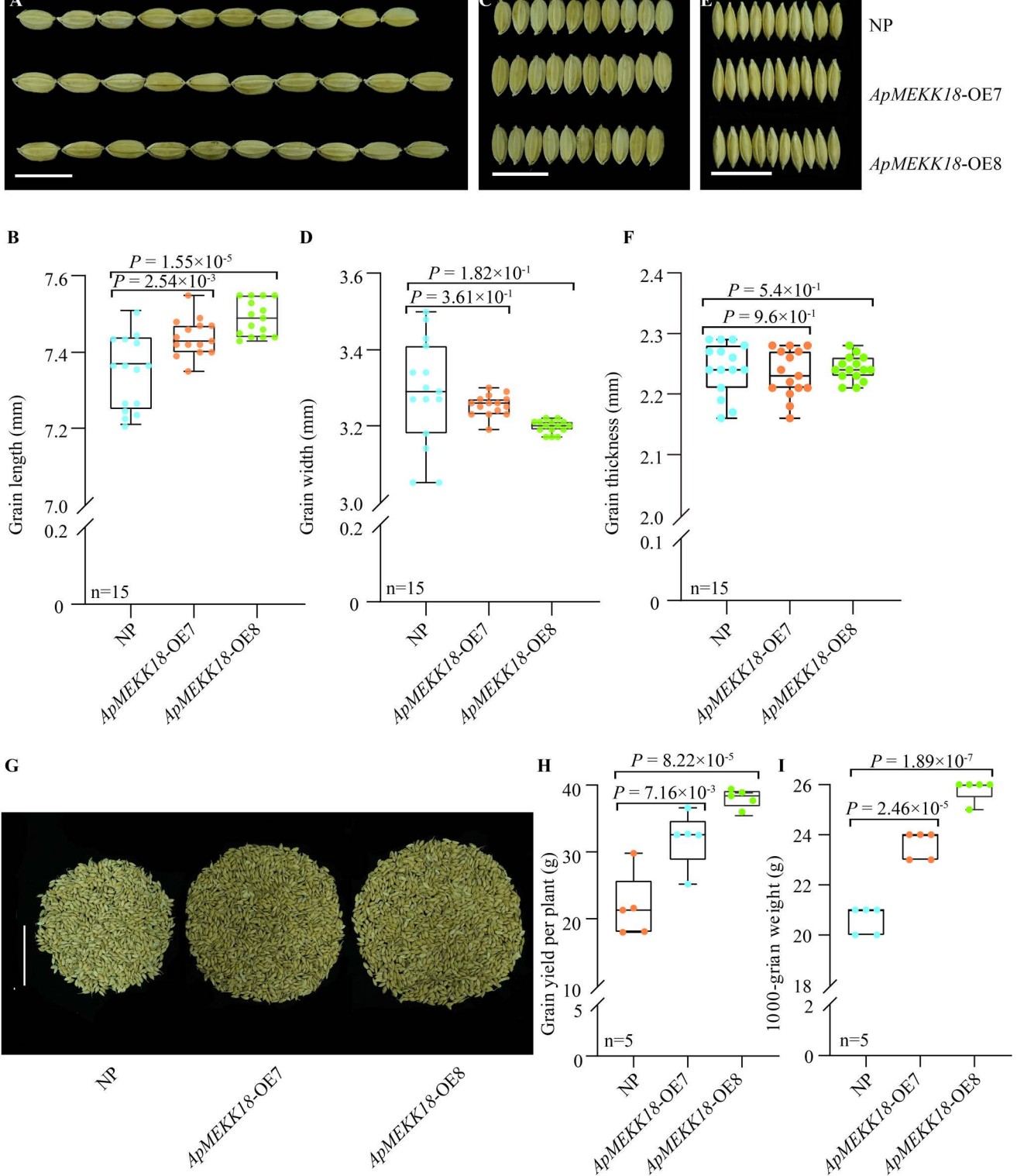

**Fig 4. Effect of *ApMEKK18* overexpression on rice grain size. (A, B)** Comparison of grain length between Nipponbare (NP) and two independent *ApMEKK18* transgenic rice lines (*ApMEKK18*-OE7 and *ApMEKK18*-OE8). **(C, D)** Comparison of grain width. **(E, F)** Comparison of grain thickness. Scale bars: 1 cm. **(G, H)** Comparison of yield per plant; scale bar: 5 cm. **(I)** Comparison of 1,000-grain weight.

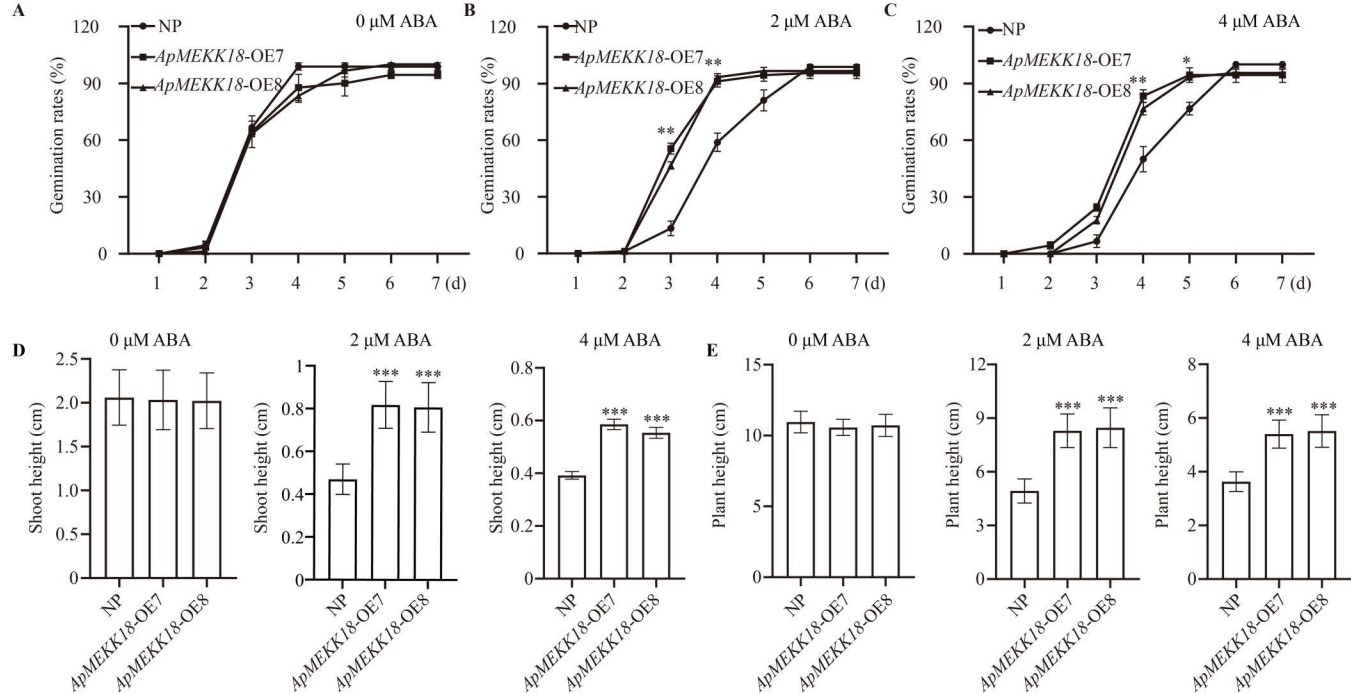

**Fig 5. Germination rate and seedling growth in rice lines overexpressing _ApMEKK18_ following treatment with different concentrations of ABA.** Rice seed germination in a hydroponic solution at 28°C containing no ABA **(A)**, 2 μM ABA **(B)**, or 4 μM ABA **(C)**. **(D)** Shoot length after 7 days of growth under treatment with 0 μM, 2 μM or 4 μM ABA. (E) Seedling height after 14 d of growth under treatment with 0 μM, 2 μM or 4 μM ABA. NP: Nipponbare; OE-7 and OE-8 represent two independent _ApMEKK18_ transgenic rice lines; one-way ANOVA (Student's _t_-test), *differences were significant ($P < 0.05$), **differences were highly significant ($P < 0.01$).

2 μM ABA, most _ApMEKK18_-OE7/OE8 seeds had germinated after 4 days, while NP seeds took 6 days to germinate fully (Fig 5B). Under 4 μM ABA, germination of NP was further delayed compared to _ApMEKK18_-OE7/OE8, with significant differences in germination rates on days 4 and 5 (Fig 5C). Overexpression of _ApMEKK18_, therefore, alleviated the inhibitory effect of ABA on rice seed germination.

Seeds germinated under similar conditions were transplanted into nutrient solutions containing different concentrations of ABA, and growth was observed after 7 and 14 days. After 7 days of treatment, under normal conditions (0 μM ABA), no statistically significant differences in shoot length were seen between _ApMEKK18_-OE7/OE8 and NP (Fig 5D and Fig 6A Fig). The application of 2 μM or 4 μM ABA inhibited the growth of _ApMEKK18_-OE7/OE8 and NP seedlings (Fig 6A Fig), but _ApMEKK18_-OE7/OE8 shoots were longer, indicating that the OE lines were less inhibited (Fig 5D). After 14 days of cultivation, under normal growth conditions, the growth of _ApMEKK18_-OE7/OE8 seedlings was consistent with that of NP, with no significant differences in plant height. However, under 2 μM and 4 μM ABA treatments, NP seedlings were significantly shorter than _ApMEKK18_-OE7/OE8 seedlings, with the effect being more severe under 4 μM ABA (Fig 5E, Fig 6B and 6C). Furthermore, we next analyzed the expression levels of genes involved the ABA signaling pathway [46]. qRT-PCR results revealed that compared with NP seedlings, the expression levels of _9-Cis-Epoxycarotenoid Dioxygenase_ (_OsNCED1_) and _OsNCED4_ were significantly reduced in _ApMEKK18_-OE7/8 seedlings, but _OsNCED2_ and _ABA 8'-Hydroxylase_ (_OsABA8ox1_) were up-regulated in _ApMEKK18_-OE7/8 seedling (S6 Fig). In summary, compared to NP, _ApMEKK18_-OE7/OE8 showed reduced sensitivity to ABA.

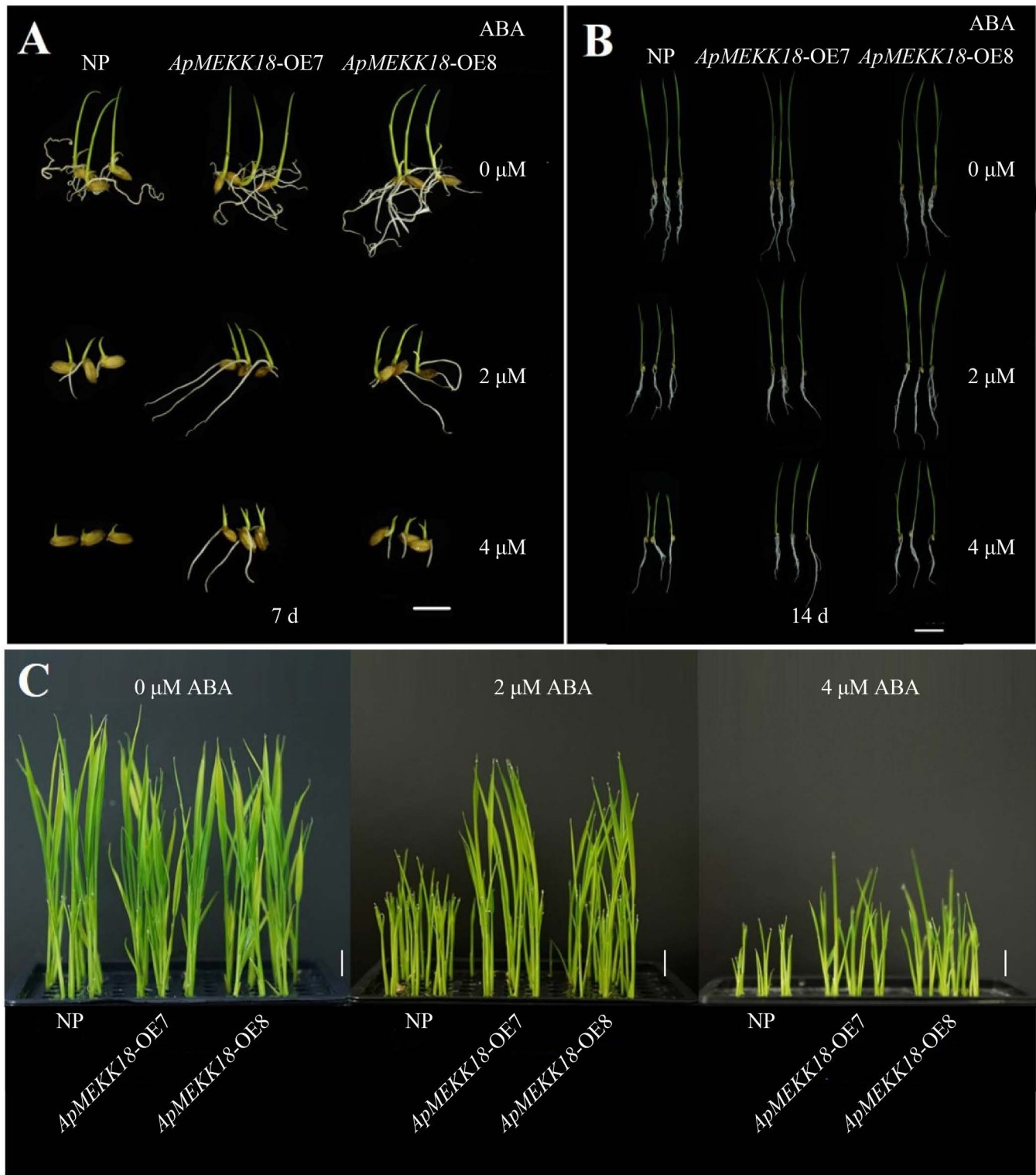

**Fig 6. Phenotypes of NP and *ApMEKK18-OE* lines treated with different concentration of ABA for 7 or 14 d. (A)** Treatment for 7 d. Scale bar: 1 cm. **(B)** Treatment for 14 d. NP: Nipponbare; OE-7 and OE-8 represent two independent *ApMEKK18* transgenic rice lines. Scale bar: 3 cm.

## Overexpression of *ApMEKK18* enhances salt stress tolerance in rice

To validate the effect of overexpressing the *ApMEKK18* gene on the salt stress response, seeds from NP and *ApMEKK18*-OE7/OE8 were germinated in 0, 50, 100, and 150 mM NaCl solutions, and the germination rates were recorded over the next 7 days. The results revealed that under standard conditions (0 mM NaCl), the germination rates of *ApMEKK18*-OE7/OE8 seeds were not significantly different from those of NP (Fig 7A). Under the 50 mM NaCl treatment, the germination rate of *ApMEKK18*-OE7/OE8 transgenic lines was significantly faster than that of NP. All transgenic seeds had germinated by day 5, while NP seeds did not fully germinate until day 6 (Fig 7B). The 100 mM NaCl treatment inhibited the germination of both *ApMEKK18*-OE7/OE8 and NP seeds, but the germination rate of NP was significantly slower than that of the transgenic lines between days 3 and 6 (Fig 7C). Under 150 mM NaCl, the *ApMEKK18*-OE7/OE8 and NP seed germination was severely inhibited, but the transgenic lines exhibited a significantly higher germination rate than NP after 3 days (Fig 7D).

Rice seedlings grown for 30 days under normal conditions (Fig 7E) were transplanted into a nutrient solution containing 150 mM NaCl for 14 days. Under standard growth conditions, there was no significant difference in the number of primary branches displayed by *ApMEKK18*-OE7/OE8 and NP seedlings, but salt stress inhibited the growth of both NP and the transgenic lines. However, NP seedlings showed more severe wilting and yellowing than *ApMEKK18*-OE7/OE8, indicating that the transgenic lines exhibited increased tolerance to salt stress (Fig 7F). After 7 days of rehydration, most leaves on the NP plants remained wilted, yellowed, and curled (Fig 6G), while the survival rate of the *ApMEKK18*-OE7/OE8 lines was appreciably higher, with all plants remaining vigorous (S7 Fig, S2Table). These results suggested that overexpression of the *ApMEKK18* gene increases salt tolerance in rice plants.

## Discussion

Drought, salinity, and other abiotic factors substantially affect agricultural production. Plants typically respond and adapt to these stresses through a range of physiological and biochemical processes. Understanding the molecular mechanisms that underlie crop responses to non-biological stresses holds great theoretical and practical significance [9].

In this study, *AtMEKK18* was induced quickly by ABA, salt, and drought stresses [31]. Mannitol-induced drought stress greatly impacted *ApMEKK18* expression, suggesting that the gene contributes significantly to the drought stress response. ABA is an essential stress hormone in plants, and the protein kinase SnRK2 is crucial to the ABA receptor core signaling pathway. Under abiotic stress, PP2C family phosphatases bind to PYR/PYL/RCARs proteins, forming a complex that binds to and inhibits PP2C [47]. SnRK2 is then released from a state of inhibition and phosphorylated, activating the ABA signaling pathway, thus initiating stress response processes [47].

In *Arabidopsis*, *AtMEKK18* is involved in seed germination, root and stomatal development, and leaf senescence. ABA affects the expression of *AtMEKK18* in specific tissues, such as the root meristem [25,32]. In this study, *ApMEKK18* exhibited different expression patterns in 13 distinct *A. pumila* tissue types, displaying higher expression levels in flowers, pedicels, buds, and young roots (S2 Fig). This expression pattern suggested that *ApMEKK18* is crucial to the growth and development of *A. pumila*, regulating specific functions in different tissues. Interestingly, this study found that ApMEKK18 was mainly located in the nucleus (Fig 2). There are many *MAPKKK* gene family members in plants. The subcellular localizations of MAPKKK proteins in plants are jointly regulated by signal stimuli, protein interactions and the characteristics of family members. Their dynamic changes are an important basis for accurate signal transduction. Different members may present different characteristics of subcellular localization due to differences in N-terminal regulatory domains and their interacting proteins [48]. For example, *Arabidopsis* AtMEKK18 is mainly localized in the nucleus and plays an important role in the ABA signaling pathway [49].

The MAPK cascade plays a significant role in ABA signal transduction, and ABA can induce several *MEKK* genes [27]. ABA induces *Arabidopsis AtMAP3Kδ4* and enhances salt tolerance. Moreover, transgenic plants overexpressing *AtMAP3Kδ4* show more robust growth, indicating that *AtMAP3Kδ4* expression is critical to the salt stress response. ABA

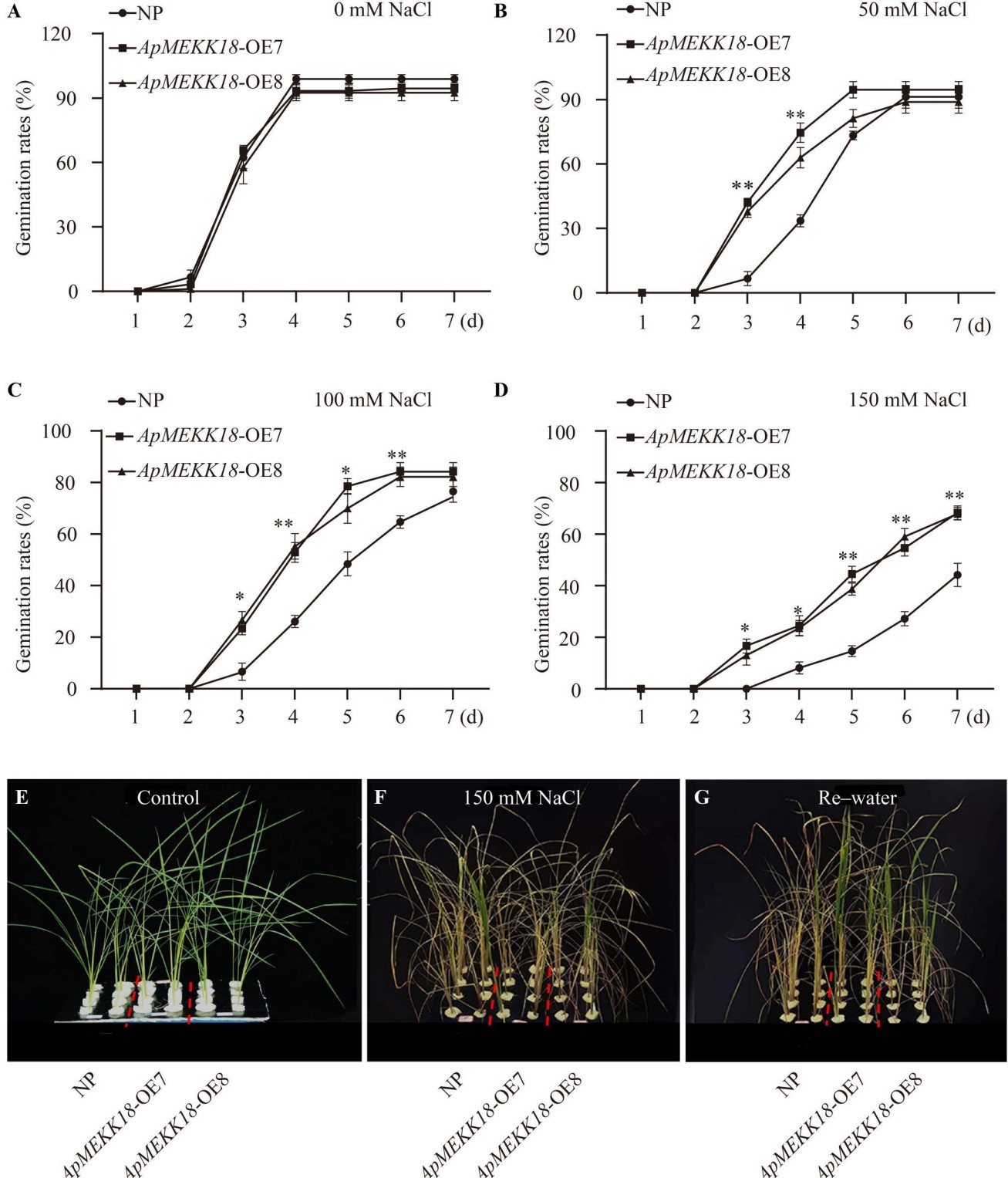

**Fig 7. Germination rate and seedling growth in rice overexpressing *ApMEKK18* treated with different concentrations of NaCl.** Rice seed germination in a hydroponic solution at 28°C containing 0 mM NaCl **(A)**, 50 mM NaCl **(B)**, 100 mM NaCl **(C)** and 150 mM NaCl **(D)**. **(E)** Phenotypes of NP and *ApMEKK18*-OE lines grown for 30 days; **(F)** phenotypes of NP and *ApMEKK18*-OE lines treated with 150 mM NaCl for 14 d; and **(G)** phenotypes

of NP and *ApMEKK18*-OE lines recovered from 150 mM NaCl treatment by 7 days of rehydration treatment. NP: Nipponbare; OE-7 and OE-8 represent two independent *ApMEKK18* transgenic rice lines; one-way ANOVA (Student's *t*-test), *differences were significant ($P < 0.05$), **differences were highly significant ($P < 0.01$).

regulates the transcriptional activation of *MEKK* genes through the PYR/PYL/RCAR-SnRK2-PP2C pathway, thus modulating *Arabidopsis* leaf senescence and the drought response [25,31,32]. While the SnRK2 protein kinase and PP2C phosphatase form the core ABA pathway, the MAPK pathway also participates in plant responses to ABA, with ABI1 regulating the activity and stimulating the proteasomal breakdown of *MEKK18* [49]. In this study, treatment with 0, 2, and 4 µM ABA led to different trends in rice seed germination (Figs 5A-C). After 7 and 14 days of treatment, shoot growth in *ApMEKK18*-OE7/OE8 was less inhibited than in NP, suggesting reduced sensitivity of transgenic lines to exogenous ABA (Figs 5D-5E). Moreover, we found that two genes involved ABA synthesis pathway [46], *OsNCED1/4* were obviously down-regulated, whereas *OsNCED2* and the *OsABA80x1*, an ABA metabolism gene, was significantly up-regulated (S6 Fig). This result suggested that overexpression of *ApMEKK18* may affect ABA synthesis or decomposition, thereby altering ABA homeostasis and reducing the sensitivity of rice seedlings to ABA. Future efforts would focus on studying how overexpression of *ApMEKK18* fine-tunes the content of ABA to regulate the growth of rice seedlings. These findings implied that the regulation of *ApMEKK18* expression is contingent upon other members of the ABA core pathway and that *ApMEKK18* significantly impacts signaling transduction, positively regulating plant tolerance to ABA. The *ApMEKK18* gene may be engaged in abiotic stress responses, working in concert with the ABA core pathway to enable plants to thrive in unfavorable environments [32].

Under salt stress at concentrations of 0, 50, 100, and 150 mM, overexpression of the *ApMEKK18* gene increased the germination rate of rice seeds (Figs 6A-6D). Under 150 mM salt stress, *ApMEKK18*-OE7/OE8 seedlings showed less severe wilting and yellowing compared to NP seedlings, and after rehydration, the two overexpression lines partially regained vitality (Fig 6E), with the survival rate of *ApMEKK18*-OE7/OE8 lines significantly higher than that of NP (S7 Fig), suggesting that *ApMEKK18* is vital to salt tolerance [10]. In this study, overexpressing *ApMEKK18* in rice enhanced salt tolerance, and it is speculated that its salt tolerance mechanism may be linked to the ABA core pathway.

Compared to NP, there were no notable differences in plant height, primary and secondary branch numbers, grain width, or grain thickness in rice *ApMEKK18* overexpression lines (Figs 3 and 4). However, the transgenic lines displayed increases in tiller number, panicle length, grain length, 1,000-grain weight, and single-plant yield (Figs 3 and 4). These findings suggested that the single-plant yield may be improved by increases in grain length and tiller number in transgenic rice lines. MEKK is an upstream regulatory factor of the MAPK cascade signaling pathway, which regulates cell division and differentiation, participates in plant growth and development, stress response, and metabolic regulation by activating MKK and MAPK [22–26]. In rice, overexpression of *OsMEKK17* regulated ABA stress response, optimized stomatal development and water use efficiency, improved drought tolerance, significantly enhanced plant adaptation to drought, and maintained high yields under drought conditions [50]. *ZmMEKK* indirectly enhanced drought resistance and yield potential in maize by promoting root development and optimizing root shoot ratio, enhancing water absorption capacity [51]. However, the molecular mechanisms underlying the regulation of rice grain length and tiller number require further investigation.

In conclusion, the *ApMEKK18* gene was found to respond significantly to ABA, NaCl, and drought stresses and the ApMEKK18 protein was shown to be localized in the nucleus. In transgenic rice, overexpressing *ApMEKK18* significantly increased single-plant yield and enhanced resistance to ABA and NaCl stresses. These outcomes demonstrated that *ApMEKK18* has a significant impact on plant stress responses, providing insights that will be useful to further exploring the function and molecular mechanisms of *ApMEKK18* in abiotic stress responses.

## Supporting information

**S1 Fig. A neighbour-joining tree based on amino acid sequeces of ApMEKK18 and selected MEKK18 sequences from *Arabidopsis* and rice.**
(TIF)

**S2 Fig. Analysis of the expression patterns of *ApMEKK18* in 13 tissues of *Arabidopsis pumila* using qRT-PCR.**
(TIF)

**S3 Fig. The genetic transformation process of rice mediated by *Agrobacterium tumefaciens* EHA105.** (A) Inducing callus tissue; (B) Succession cultivation; (C) Cocultivation; (D) Choose cultivation; (E) Differentiation cultivation; (F) Rooting; (G) Transplant.
(TIF)

**S4 Fig. PCR detection of T$_3$ generation rice plants overexpressing *ApMEKK18* gene.** Maker: DL5000; +: Positive control; -: Negative control; NP: Nipponbare; OE-1～9 represent nine independent *ApMEKK18* transgenic rice lines. The upper half of the electrophoresis gel was not related to this study.
(TIF)

**S5 Fig. Analysis of the expression level of *ApMEKK18* in Nipponbare and *ApMEKK18* transgenic lines using qRT-PCR.** NP: Nipponbare; OE-1～9 represent independent *ApMEKK18* transgenic rice lines; one-way ANOVA (Student's *t*-test), *differences were significant ($P < 0.05$), **differences were highly significant ($P < 0.01$).
(TIF)

**S6 Fig. Rice endogenous ABA signaling pathway gene expression patterns in the wild-type (NP) and *ApMEKK18-OE* seedlings.** *OsNCED1*, *OsNCED2*, and *OsNCED4*: 9-Cis-Epoxycarotenoid Dioxygenase; *OsABA8ox1*, *ABA 8′-Hydroxylase*. *differences were significant ($P < 0.05$), **differences were highly significant ($P < 0.01$).
(TIF)

**S7 Fig. Survival rate of rice seedlings overexpressing *ApMEKK18* gene in NP after 14 d of 150 mmol/L stress treatment followed by 7 d of rehydration.** NP: Nipponbare; OE-7 and OE-8 represent two independent *ApMEKK18* transgenic rice lines; one-way ANOVA (Student's *t*-test), * $P < 0.05$, ** $P < 0.01$.
(TIF)

**S1 Table. Primers used in this study.**
(XLSX)

**S2 Table. Survival rate of rice seedling after rehydration for 7 days under 150 mM NaCl stress.**
(XLSX)

## Author contributions

**Conceptualization:** Xianzhong Huang.

**Data curation:** Junrong Li, Xianzhong Huang.

**Formal analysis:** Junrong Li.

**Funding acquisition:** Xianzhong Huang.

**Investigation:** Zhanglun Sun, Edwin Afriyie Owusu, Leiming Liu, Tianrun Mei.

**Methodology:** Yilin Yang, Xianzhong Huang.

**Supervision:** Xianzhong Huang.

**Validation:** Zhanglun Sun, Qingli Zhang.

**Writing – original draft:** Junrong Li, Yilin Yang, Zhanglun Sun.

**Writing – review & editing:** Xianzhong Huang.

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
