## [Decision Letter · Decision Letter 0]

Dear Dr. Huang,

Thank you for submitting your manuscript to PLOS ONE. After careful consideration, we feel that it has merit but does not fully meet PLOS ONE’s publication criteria as it currently stands. Therefore, we invite you to submit a revised version of the manuscript that addresses the points raised during the review process.

I have now received feedback from the reviewers. The reviewers have raised some very relevant points that need to be addressed before a decision can be made on the manuscript.

I request you to respond to the reviewers comments and revise the manuscript accordingly. Please ensure that all concerns are adequately addressed in your response.

. ==============================

We look forward to receiving your revised manuscript.

Kind regards,

Koppolu Raja Rajesh Kumar, PhD

Academic Editor

PLOS ONE

 [This work was supported by the National Natural Science Foundation of China (32270385), the Excellent Scientific Research and Innovation team of the Education Department of Anhui Province (2022AH010087), the Science and technology innovation team of Anhui Sciences and Technology University (2023KJCXTD001), the Talent Introduction Start-up Fund Project of Anhui Science and Technology University (NXYJ202001), the Construction Funds for Crop Science of Anhui Science and Technology University (XK-XJGF001), and the National Innovation and Entrepreneurship Training Program for College Students (202310879009).]. 

[This work was supported by the National Natural Science Foundation of China (32270385), the Excellent Scientific Research and Innovation team of the Education Department of Anhui Province (2022AH010087), the Science and technology innovation team of Anhui Sciences and Technology University (2023KJCXTD001), the Talent Introduction Start-up Fund Project of Anhui Science and Technology University (NXYJ202001), the Construction Funds for Crop Science of Anhui Science and Technology University (XK-XJGF001), and the National Innovation and Entrepreneurship Training Program for College Students (202310879009).]. 

[This work was supported by the National Natural Science Foundation of China (32270385), the Excellent Scientific Research and Innovation team of the Education Department of Anhui Province (2022AH010087), the Science and technology innovation team of Anhui Sciences and Technology University (2023KJCXTD001), the Talent Introduction Start-up Fund Project of Anhui Science and Technology University (NXYJ202001), the Construction Funds for Crop Science of Anhui Science and Technology University (XK-XJGF001), and the National Innovation and Entrepreneurship Training Program for College Students (202310879009).]. 

Reviewers' comments:

Reviewer's Responses to Questions

**Comments to the Author**

1. Is the manuscript technically sound, and do the data support the conclusions?

Reviewer #1: Partly

Reviewer #2: Yes

2. Has the statistical analysis been performed appropriately and rigorously?

Reviewer #1: Yes

Reviewer #2: Yes

3. Have the authors made all data underlying the findings in their manuscript fully available?

Reviewer #1: Yes

Reviewer #2: Yes

4. Is the manuscript presented in an intelligible fashion and written in standard English?

Reviewer #1: Yes

Reviewer #2: Yes

Reviewer #1: This manuscript describes the overexpression of ApMEKK18 from Arabidopsis pumila increases tolerances to osmotic, salinity, and ABA treatment. The work is OK and informative. However, I have several concerns requiring a revision before this manuscript can be accepted.

(1) MEKK is a large gene family in plants. Does rice have its own MEKK18? Please add a phylogenetic analysis for ApMEKK18 in Arabidopsis pumila, Arabidopsis thaliana, and rice.

(2) Please provide the statistical data such as survival ratio or ratio of dead leaves for Figure 6E-G.

(3) The author found that the ApMEKK18 was mainly located in nucleus. However, MAPKKK (MEKK) is commonly the first step of MAPK pathway. How it works in nucleus. Authors should discuss this.

(4) Authors suggested the ApMEKK18 enhanced tolerances by through ABA pathway. Please test the expression of some genes associated with ABA metabolism and signal transduction in WT and overexpression lines.

(5) As the most stress tolerances were measured in germination and seedling stages, I recommend the title as “Overexpression of MEKK18 from Arabidopsis pumila in rice significantly enhances stress resistance at the early stage”.

(6) How ApMEKK18 enhances the productivity in rice? This is not clear. Please add some sentences to discuss this.

Reviewer #2: As the authors emphasized in Abstract and conclusion, the results of this study provide a basis for further research on the function of ApMEKK18 in stress response. To improve the implication of current study, the phenotype of the transgenic rice plants under drought stress should be analyzed and shown.

To understand at least a hint behind the role of ApMEKK18 in seed yield and/or stress tolerance, the potential target genes involved in ABA, salt, and drought response should be analyzed.

Figure 1; the expression patterns of MpMEKK18 seem to follow circadian rhythm. To rule out the circadian effects, its expression level should be measured at different time points under normal conditions. In Figure legend, the developmental stage of the plant material used should be described.

Figure 2; “Chlorophyll indicated by chloroplast autofluorescence” in figure legend is wrong; the red signals should be chlorophyll autofluorescence.

Figure S5 is an important supporting data, which should be moved to main Figure 5.

Figure 6; only germination rates were measured under salt stress conditions. To further support the role of MpMEKK18 in salt tolerance, other parameters, such as survival rate and chlorophyll content, should be measured.

Line 265-265; ApMEKK18 gene expression levels in 13 tissue types from A. pumila were analyzed using qRT-PCR. Flowers exhibited the highest expression, followed by the pedicel, flower buds, and young roots (S1 Fig), which is wrong. Figure S4 should be S1. Subsequent Supplementary figure numbers are also wrong.

**Do you want your identity to be public for this peer review?** For information about this choice, including consent withdrawal, please see our Privacy Policy

Reviewer #1: No

Reviewer #2: No

---

## [Author Response · Author response to Decision Letter 1]

14 Apr 2025

Dear editors,

Thank you very much for taking the time to review our manuscript entitled “Overexpression of MEKK18 from Arabidopsis pumila in rice significantly enhances stress resistance” (Submission ID: PONE-D-25-02616). We sincerely appreciate the reviewers for their valuable comments and suggestions on our manuscript. These comments are very helpful in revising and improving our manuscript, and providing important guidance for our future research work.

According to the comments of editors and reviews, we have made a careful correction in the revised manuscript. We hope this meets the requirements for a publication. All the revised sections of the text were highlighted using track changes. Our specific responses to each question are as follows:

We ensure that we provided the correct grant numbers, and amended the ‘Acknowledgments section’ to ‘Funding Information’, and we confirm that the funders had no role in study design, data collection and analysis, decision to publish, or preparation of the manuscript. Also, we provided the original uncropped and unadjusted images of S4 Fig in the revised manuscript.

Reviewer #1: This manuscript describes the overexpression of ApMEKK18 from Arabidopsis pumila increases tolerances to osmotic, salinity, and ABA treatment. The work is OK and informative. However, I have several concerns requiring a revision before this manuscript can be accepted.

(1) MEKK is a large gene family in plants. Does rice have its own MEKK18? Please add a phylogenetic analysis for ApMEKK18 in Arabidopsis pumila, Arabidopsis thaliana, and rice.

Response: Thank you for your suggestion. According to your suggestion, we used the amino acid sequence of ApMEKK18 as a query to blast in the rice database, downloaded the OsMEKK18 partial homologous protein of rice together with the AtMEKK18 partial homologous protein reported in Arabidopsis, and constructed a phylogenetic tree of 11 MEKK18 proteins, as a supplementary figure S1 Fig. Accordingly, the original S1 Fig was renamed S2 fig, and so on. The results showed that ApMEKK18 was closest to AtMEKK18, but farthest from OsMEKK18.

(2) Please provide the statistical data such as survival ratio or ratio of dead leaves for Figure 6E-G.

Response: Thanks! We provided the detailed statistical data on the survival rate of rice seedlings in Figure 6E-D. The corresponding statistical data can be found in Table S2.

(3) The author found that the ApMEKK18 was mainly located in nucleus. However, MAPKKK (MEKK) is commonly the first step of MAPK pathway. How it works in nucleus. Authors should discuss this.

Response: Thank you for your very good reviews! There are many MAPKKK gene family members in plants. The subcellular localizations of MAPKKK proteins in plants are jointly regulated by signal stimuli, protein interactions and the characteristics of family members. Their dynamic changes are an important basis for accurate signal transduction. Different members may present different characteristics of subcellular localization due to differences in N-terminal regulatory domains and their interacting proteins (Tajdel-Zielińska et al., 2024). For example, Arabidopsis AtMEKK18 is mainly localized in the nucleus and plays an important role in the ABA signaling pathway (Mitula et al., 2015). In the discussion part of the revised manuscript, relevant discussions are supplemented.

The references are as follows:

[48] Tajdel-Zielińska M, Janicki M, Marczak M, Ludwików A. Arabidopsis HECT and RING-type E3 Ligases Promote MAPKKK18 Degradation to Regulate Abscisic Acid Signaling. Plant & Cell Physiology. 2024; 65(3): 390-404. https://doi.org/ 10.1093/pcp/pcad165. PMID: 38153765.

[49] Mitula F, Tajdel M, Cieśla A, Kasprowicz-Maluśki A, Kulik A, Babula-Skowrońska D, Michalak M, Dobrowolska G, Sadowski J, Ludwików A. Arabidopsis ABA-Activated Kinase MAPKKK18 is Regulated by Protein Phosphatase 2C ABI1 and the Ubiquitin-Proteasome Pathway. Plant & Cell Physiology. 2015; 56(12): 2351-2367. https://doi.org/10.1093/pcp/pcv146. PMID: 26443375.

(4) Authors suggested the ApMEKK18 enhanced tolerances by through ABA pathway. Please test the expression of some genes associated with ABA metabolism and signal transduction in WT and overexpression lines.

Response: Thank you for your suggestion. We analyzed the expression levels of three ABA synthesis genes, 9-Cis-Epoxycarotenoid Dioxygenase (OsNCED1), OsNCED2, and OsNCED4, and one ABA metabolism gene, ABA 8′-Hydroxylase (OsABA8ox1) in the control (NP) and ApMEKK18-OE8 plants. qRT PCR results showed that compared with NP seedlings, the expression levels of OsNCED1 and OsNCED4 genes were significantly reduced in ApMEKK18-OE8 seedlings, but OsNCED2 was up-regulated in ApMEKK18-OE8 seedling, while OsABA8ox1 was significantly down regulated in ApMEKK18-OE8 (S6 Fig.).

(5) As the most stress tolerances were measured in germination and seedling stages, I recommend the title as “Overexpression of MEKK18 from Arabidopsis pumila in rice significantly enhances stress resistance at the early stage”.

Response: Thanks. According to your suggestion, we amended the title to “Overexpression of MEKK18 from Arabidopsis pumila in rice significantly enhances stress resistance at the early stage”.

(6) How ApMEKK18 enhances the productivity in rice? This is not clear. Please add some sentences to discuss this.

Response: Thanks. MEKK is an upstream regulatory factor of the MAPK cascade signaling pathway, which regulates cell division and differentiation, participates in plant growth and development, stress response, and metabolic regulation by activating MKK and MAPK. In rice, overexpression of OsMEKK17 gene regulated ABA stress response, optimized stomatal development and water use efficiency, improved drought tolerance, significantly enhanced plant adaptation to drought, and maintained high yields under drought conditions (Cui et al., 2024). ZmMEKK indirectly enhanced drought resistance and yield potential in maize by promoting root development and optimizing root shoot ratio, enhancing water absorption capacity (Shi et al., 2022). In the discussion of the revised manuscript, relevant discussions were added.

The references are as follows:

[50] Cui L, Song Y, Zhao Y, Gao R, Wang Y, Lin Q, Jiang J, Xie H, Cai Q, Zhu Y, Xie H, Zhang J. Nei 6 You 7075, a hybrid rice cultivar, exhibits enhanced disease resistance and drought tolerance traits. BMC Plant Biology. 2024; 24(1): 1252. https://doi.org/10.1186/s12870-024-05998-2. PMID: 39725902; PMCID: PMC11670435.

[51] Shi Z, Zhao B, Song W, Liu Y, Zhou M, Wang J, Zhao J, Ren W. Genome-wide identification and characterization of the MAPKKK, MKK, and MPK families in Chinese elite maize inbred line Huangzaosi. Plant Genome. 2022; 15(3): e20216. https://doi.org/10.1002/tpg2.20216. PMID: 35535627.

Reviewer #2: As the authors emphasized in Abstract and conclusion, the results of this study provide a basis for further research on the function of ApMEKK18 in stress response. To improve the implication of current study, the phenotype of the transgenic rice plants under drought stress should be analyzed and shown.

To understand at least a hint behind the role of ApMEKK18 in seed yield and/or stress tolerance, the potential target genes involved in ABA, salt, and drought response should be analyzed.

Figure 1; the expression patterns of MpMEKK18 seem to follow circadian rhythm. To rule out the circadian effects, its expression level should be measured at different time points under normal conditions. In Figure legend, the developmental stage of the plant material used should be described.

Response: Thank you for your insightful reviews. Figure 1 showed the expression profiles of the ApMEKK18 gene under four abiotic stresses within 12 hours. Gene expression exhibited different characteristics of change, and the expression pattern did not follow the regulated expression pattern. Moreover, the samples were all grown under sunlight conditions. Our previous research found that during 48 hours of continuous high salt stress, the expression of the ApMEKK18 gene continued to upregulate, without showing a rhythmic pattern (Yang et al., 2018). In the caption, we have added four time point descriptions.

Yang LF, Jin YH, Huang W, Sun Q, Liu F, Huang XZ. Full-length transcriptome sequences of ephemeral plant Arabidopsis pumila provides insight into gene expression dynamics during continuous salt stress. BMC Genomics. 2018; 19(1): 717. doi: 10.1186/s12864-018-5106-y. PMID: 30261913; PMCID: PMC6161380.

Figure 2; “Chlorophyll indicated by chloroplast autofluorescence” in figure legend is wrong; the red signals should be chlorophyll autofluorescence.

Response: Thanks. We apologized for the mistake and corrected it in the revised manuscript.

Figure S5 is an important supporting data, which should be moved to main Figure 5.

Response: Thanks. According to your suggestion, S5 Fig. was moved to main text as Fig 6 and the corresponding description was modified. Correspondingly, Fig 6 was renamed Fig 7.

Figure 6; only germination rates were measured under salt stress conditions. To further support the role of MpMEKK18 in salt tolerance, other parameters, such as survival rate and chlorophyll content, should be measured.

Response: Thanks. Based on your and Reviewer 1's suggestions, in the revised manuscript we added the statistical data on survival rates (S2 Table).

Line 265-265; ApMEKK18 gene expression levels in 13 tissue types from A. pumila were analyzed using qRT-PCR. Flowers exhibited the highest expression, followed by the pedicel, flower buds, and young roots (S1 Fig), which is wrong. Figure S4 should be S1. Subsequent Supplementary figure numbers are also wrong.

Response: Thank you for your careful reviews. We carefully checked the numbering of the supplementary figures and made corrections to ensure accurate naming of all the figures.

---

## [Decision Letter · Decision Letter 1]

Dear Dr. Huang,

Thank you for submitting your manuscript to PLOS ONE. After careful consideration, we feel that it has merit but does not fully meet PLOS ONE’s publication criteria as it currently stands. Therefore, we invite you to submit a revised version of the manuscript that addresses the points raised during the review process.

The reviewers have evaluated your revised manuscript. I am pleased to inform you that the manuscript can be accepted for publication once the comments from Reviewer 2 are fully addressed and incorporated into the manuscript.

I look forward to receiving your revised version.

We look forward to receiving your revised manuscript.

Kind regards,

Koppolu Raja Rajesh Kumar, PhD

Academic Editor

PLOS ONE

Journal Requirements:

Reviewers' comments:

Reviewer's Responses to Questions

**Comments to the Author**

Reviewer #1: All comments have been addressed

Reviewer #2: (No Response)

2. Is the manuscript technically sound, and do the data support the conclusions?

Reviewer #1: Yes

Reviewer #2: Yes

3. Has the statistical analysis been performed appropriately and rigorously?

Reviewer #1: Yes

Reviewer #2: Yes

4. Have the authors made all data underlying the findings in their manuscript fully available?

Reviewer #1: Yes

Reviewer #2: Yes

5. Is the manuscript presented in an intelligible fashion and written in standard English?

Reviewer #1: Yes

Reviewer #2: Yes

Reviewer #1: Good job! This manuscript is much improved since the last revision. I am satisfied with it and have no further concerns.

Reviewer #2: The authors addressed all of my comments and suggestions with additional experimental data, which improves and clarifies the manuscript. However, a couple of points should be further considered. Fig. S6 showing the expression levels of ABA-related genes is crucial for supporting the role of ApMEKK18 in ABA response. First, the description in line 368-369 “while ABA 8′-Hydroxylase (OsABA8ox1) was significantly down regulated in ApMEKK18-OE8 (S6 Fig)” is wrong; its level was upregulated in the transgenic lines. Moreover, it is necessary to analyze the expression levels of these genes in other transgenic line (OE 7 line) to further support the conclusion. In addition, it should be discussed in Discussion section that how the up- and down-regulation of these genes is associated with the ABA-responsive phenotype of the plants.

**Do you want your identity to be public for this peer review?** For information about this choice, including consent withdrawal, please see our Privacy Policy

Reviewer #1: No

Reviewer #2: No

---

## [Author Response · Author response to Decision Letter 2]

12 May 2025

Dear editors,

Thank you very much for taking the time to review our manuscript entitled “Overexpression of MEKK18 from Arabidopsis pumila in rice significantly enhances stress resistance” (Submission ID: PONE-D-25-02616) again.

According to the comments of Reviewer #2, we have made a careful correction in the revised manuscript. We hope this meets the requirements for a publication. All the revised sections of the text were highlighted using track changes. Our specific responses to each question are as follows:

We ensure that our references are complete and correct, and we have not cited papers that have been retracted.

Reviewer #2: The authors addressed all of my comments and suggestions with additional experimental data, which improves and clarifies the manuscript. However, a couple of points should be further considered. Fig. S6 showing the expression levels of ABA-related genes is crucial for supporting the role of ApMEKK18 in ABA response. First, the description in line 368-369 “while ABA 8′-Hydroxylase (OsABA8ox1) was significantly down regulated in ApMEKK18-OE8 (S6 Fig)” is wrong; its level was upregulated in the transgenic lines. Moreover, it is necessary to analyze the expression levels of these genes in other transgenic line (OE 7 line) to further support the conclusion. In addition, it should be discussed in Discussion section that how the up- and down-regulation of these genes is associated with the ABA-responsive phenotype of the plants.

Response: Thank you for your careful reviews. We apologized for the mistake and corrected it in the revised manuscript. We have analyzed the expression levels of these genes in ApMEKK18-OE7 line, and revised the S6 Fig. in the revised manuscript. Furthermore, we added the corresponding descriptions of the implications of changes in gene expression levels in Discussion section.

---

## [Editor Report · Decision Letter 2]

Overexpression of MEKK18  from Arabidopsis pumila  in rice significantly enhances stress resistance at the early stage

PONE-D-25-02616R2

Dear Dr. Huang,

We’re pleased to inform you that your manuscript has been judged scientifically suitable for publication and will be formally accepted for publication once it meets all outstanding technical requirements.

Kind regards,

Koppolu Raja Rajesh Kumar, PhD

Academic Editor

PLOS ONE
---

## [Editor Report · Acceptance letter]

PONE-D-25-02616R2

PLOS ONE

Dear Dr. Huang,

I'm pleased to inform you that your manuscript has been deemed suitable for publication in PLOS ONE. Congratulations! Your manuscript is now being handed over to our production team.

Kind regards,

on behalf of

Dr. Koppolu Raja Rajesh Kumar

Academic Editor

PLOS ONE